# Synthesis and Na^+^ Ion Conductivity of Stoichiometric Na_3_Zr_2_Si_2_PO_12_ by Liquid-Phase Sintering with NaPO_3_ Glass

**DOI:** 10.3390/ma14143790

**Published:** 2021-07-06

**Authors:** Yongzheng Ji, Tsuyoshi Honma, Takayuki Komatsu

**Affiliations:** Department of Materials Science and Technology, Nagaoka University of Technology, Niigata 940-2188, Japan; s153427@stn.nagaokaut.ac.jp (Y.J.); komatsu@mst.nagaokaut.ac.jp (T.K.)

**Keywords:** glass-ceramics, sodium ion-batteries, solid electrolyte, liquid-phase sintering

## Abstract

Sodium super ionic conductor (NASICON)-type Na_3_Zr_2_Si_2_PO_12_ (NZSP) with the advantages of the high ionic conductivity, stability and safety is one of the most famous solid-state electrolytes. NZSP, however, requires the high sintering temperature about 1200 °C and long sintering time in the conventional solid-state reaction (SSR) method. In this study, the liquid-phase sintering (LPS) method was applied to synthesize NZSP with the use of NaPO_3_ glass with a low glass transition temperature of 292 °C. The formation of NZSP was confirmed by X-ray diffraction analyses in the samples obtained by the LPS method for the mixture of Na_2_ZrSi_2_O_7_, ZrO_2_, and NaPO_3_ glass. The sample sintered at 1000 °C for 10 h exhibited a higher Na^+^ ion conductivity of 1.81 mS/cm at 100 °C and a lower activation energy of 0.18 eV compared with the samples prepared by the SSR method. It is proposed that a new LPE method is effective for the synthesis of NZSP and the NaPO_3_ glass has a great contribution to the Na^+^ diffusion at the grain boundaries.

## 1. Introduction

All-solid-state batteries (ASSBs) have received much attention because of their high safety performance compared with the conventional lithium-ion batteries (LIBs) using liquid electrolytes [1,2,3,4]. ASSBs have been also expected to have high capacities, because cathode, solid electrolyte, anode, and collecting electrode can be integrated in ASSBs and high-density loadings are allowed. Recently, all-solid-state sodium batteries (Na-ASSBs) including no rare metals such as Li and Co have been proposed as prototypes [5,6,7,8,9]. Sodium has an energy density per weight lower than that of lithium, but this disadvantage would be largely improved in ASSBs. Yamauchi et al. [8,9] proposed a new prototype Na-ASSBs, in which Na_2_FeP_2_O_7_ glass-ceramics and β-alumina solid solutions are used as cathode active materials and solid electrolytes, respectively. Na_2_FeP_2_O_7_ glass and its derivative glass-ceramics exhibit viscous flows during the crystallization process and can be adhered to solid electrolytes at a low temperature of 500 °C without any external pressurization [10,11,12,13,14,15]. The battery’s internal resistance was successfully reduced by optimizing the crystallization process of Na_2_FeP_2_O_7_ glass, and the battery operation at −20 °C was demonstrated [9].

Solid electrolytes constituting ASSBs can be roughly classified into sulfides and oxides. The low Young’s modulus of sulfide solid electrolytes allows them to deform plastically only under pressure at room temperature, consequently suppressing the resistance against Na^+^ ion conductivity at grain boundaries [16]. Na_3_Zr_2_Si_2_PO_12_ (NZSP) reported first by Goodenough et al. [17] is a well-known oxide ionic conductor with a high Na^+^ ion conductivity of 0.2 S/cm at 300 °C. The synthesis of well-densified NZSP by classical solid-state reaction requires a long sintering process at 1250 °C to improve ionic conductivity at room temperature [18,19]. Heat treatment at high temperature is essential if ZrO_2_ is used as a raw material. A sol-gel method is also possible to synthesize NZSP by fast sintering at 1000 °C from fine xerogels as precursor, but there are issues with economical commercial production. [20] For the use of NZSP as solid electrolytes in ASSBs, the internal resistance against Na^+^ ion conductivity must be largely improved and overcome through the design of the morphology and grain boundary of NZSP. Generally, it is obvious that plastic deformation cannot be expected in oxide with high Young’s moduli.

Glasses exhibit viscous flow at temperatures above the glass transition temperature and have open structure being an advantage for ion diffusion. Glasses have been, therefore, used as sintering aids and sealants for the production of functional ceramics and devices such as fuel cells and bio-active ceramics [21,22,23,24,25,26]. Okamoto et al. [27] applied Na_2_O-Nb_2_O_5_-P_2_O_5_ glasses as sintering additives to NZSP and found that the composite obtained by the heat treatment at 900 °C for 10 min exhibits the electrical conductivity (σ) of σ = 1.2 × 10^−4^ Scm^−1^ at 25 °C. There have been several reports on the synthesis of NZSP using sintering additives [28,29,30,31,32]. The addition of sintering additives to NZSP means, however, that the chemical composition of the composite obtained deviates from the stoichiometric composition of NZSP.

In this study, we propose using a NaPO_3_ glass (50Na_2_O-50P_2_O_5_) as one component of raw materials for the synthesis of stoichiometric NZSP. Since the NaPO_3_ glass has a low glass transition temperature, it is expected that a liquid phase is created during the reaction of raw materials [33,34,35,36,37,38], i.e., the liquid-phase sintering (LPS). In other words, the formation of NZSP would be expected to occur at lower temperatures and in shorter times in the use of glass as a raw material compared with the SSR method using crystalline compounds as raw materials. This paper describes the synthesis of Na_3_Zr_2_Si_2_PO_12_ materials with the stoichiometric composition by sintering of the mixture of raw materials with the molar ratio of Na_2_ZrSi_2_O_7_:ZrO_2_:NaPO_3_ glass = 1:1:1 and their microstructure and Na^+^ ion conductivity. There has been no report on the synthesis of NZSP using NaPO_3_ glass.

## 2. Materials and Methods

The NaPO_3_ glass was prepared by the melt-quenching method. The raw material sodium dihydrogen phosphate (98.0% NaH_2_PO_4_, Nakarai Tesque Co. (Kyoto, Japan) was weighted in a platinum crucible and then pre-sintered at 500 °C for 8 h at the heating rate of 1 °C/min. The pre-sintered sample was melted at 1000 °C for 30 min in air, and the melt was quenched using a steel plate. Na_2_ZrSi_2_O_7_-ZrO_2_ composite was prepare by the SSR method. Sodium carbonate (Na_2_CO_3_), silicon dioxide (SiO_2_), and zirconium oxide (ZrO_2_) were weighted and mixed with 10 mL acetone in the condition of 15 min × 4 cycles in 700 rpm with a 10 min waiting time using a wet ball-milling (FRITSCH P-7). The mixture obtained by milling was dried at 120 °C for 12 h, and the calcination of the mixture was conducted at 900 °C for 4 h.

The calcined Na_2_ZrSi_2_O_7_-ZrO_2_ composite powder and NaPO_3_ glass were grounded manually to keep the particle size less or equal 32 µm by a sieve, and their powders were mixed mechanically for 30 min. The mixed powders were pressed into pellets with a diameter of 13 mm at a pressure of around 100 MPa. The pellets were placed on a platinum plate and sintered at 1000, 1100, and 1200 °C for 3, 5, 7, and 10 h. The temperature elevation rate to reach the target temperature was set to 10 K/min. In this article, the samples obtained using the above processing is designated as LPS-NZSP. For comparison, NZSP materials were also prepared by the conventional SSR method, in which sodium carbonate Na_2_CO_3_, sodium dihydrogen phosphate NaH_2_PO_4_, silicon dioxide SiO_2_, and zirconium oxide ZrO_2_ were used directly. The conditions of mixing by a wet ball-milling, drying, calcination, grounding, pressing, and sintering were the same as those in the synthesis of LPS-NZSP materials. The samples obtained by the SSR method is designated as SSR-NZSP.

The glass transition temperature *T*_g_, crystallization onset temperature *T*_x_, and crystallization peak temperature *T*_p_ of NaPO_3_ glass prepared in this study were determined from differential thermal analysis (DTA, Thermoplus TG-8120, RIGAKU Corp., Akishima, Tokyo, Japan). Bulk densities were calculated from the measured weight, thickness and diameter of the sintered pellets. The crystalline phases in the samples obtained by different sintering temperatures, and times were identified by X-ray diffraction (XRD) analysis (Rigaku Ultima IV X-ray diffractometer) with Cu-Kα radiation (λ = 0.154056 nm), in which the scanning speed was 5 °C/min and the diffraction angle was 2θ = 10–70°. The Rietveld analysis (PROFEX) [39] was also performed to get information on the identification and quantification of the crystalline phases in the sintered samples. The microstructure of the cross-section of the sintered samples was examined from scanning electron microscope (SEM, KEYENECE VE-8800, Osaka, Japan) observations. X-ray photoelectron spectroscopy (XPS) (ULVAC-PHI PHI5000 Versa Probe II spectrometer, Chigasaki, Kanagawa, Japan) measurements were carried out to determine the intensity of Na, Zr, Si, P, and O elements in the sintered samples, in which the samples were assembled in an Ar-filled glove box to avoid oxidation and contamination and the surface (the thickness of 0.1 mm) of the samples was polished by diamond wheel to remove contamination by carbon dioxide in the air.

Electrical conductivities, i.e., Na^+^ ion conductivities, of the sintered samples in the temperature range of 100–200 °C were measured by an alternating current (AC) impedance method using an impedance analyzer (HIOKI IM3570, Ueda, Nagano, Japan) in the frequency range of 4–5 MHz. Gold was sputtered onto both sides of the sintered samples (the thickness for 10 nm and the diameter of 6.0 mm φ) as electrodes. Activation energies for Na^+^ ion conductivity of the sintered samples were evaluated from the temperature dependence of electrical conductivity.

## 3. Results and Discussion

### 3.1. Thermal Properties of NaPO_3_ Glass

The DTA curves for the bulk and powder samples of NaPO_3_ glass at a heating rate of 10 K/min are shown in Figure 1. In the bulk sample, an endothermic dip due to the glass transition and an exothermic peak due to the crystallization are observed, indicating the values of *T*_g_ = 292 °C, *T*_x_ = 425 °C, and *T*_p_ = 425 °C for NaPO_3_ glass. A sharp endothermic peak with a strong intensity due to the melting is observed at 635 °C. On the other hand, the powder sample has the value of *T*_p_ = 328 °C, which is much lower than that in the bulk sample. This result suggests that NaPO_3_ glass prefers the surface crystallization.

In the present study, the mixture of Na_2_ZrSi_2_O_7_-ZrO_2_ composite powder and NaPO_3_ glass is sintered at 1000, 1100, and 1200 °C to synthesize NZSP. The DTA results shown in Figure 1 indicate that the crystallized NaPO_3_ glass become a liquid phase at temperatures higher than 650 °C, i.e., it is expected that Na_2_ZrSi_2_O_7_-ZrO_2_ composite powders are surrounded by a liquid phase with the chemical composition of NaPO_3_ and consequently reactions among the components of Na_2_ZrSi_2_O_7_, ZrO_2_, and NaPO_3_ at temperatures of 1000–1200 °C are largely enhanced. In this sense, the use a NaPO_3_ glass for the synthesis of stoichiometric NZSP would be regarded as the liquid-phase sintering method.

### 3.2. Formation and Microstructure of Na_3_Zr_2_Si_2_PO_12_

The XRD patterns at room temperature for the LPS-NZSP samples obtained by sintering at different temperatures (1000, 1100, and 1200 °C) and times (3–10 h) are shown in Figure 2. It is seen that the intensity of the peaks corresponding to the crystalline NZSP (Na_3_Zr_2_Si_2_PO_12_) phase increases with increasing temperature and time. The samples sintered at 1000 °C and 1200 °C for 10 h consist of mainly NZSP, although the presence of Na_2_ZrSi_2_O_7_ and ZrO_2_ phases is still detected. The results shown in Figure 2 indicate that reactions among Na_2_ZrSi_2_O_7_, ZrO_2_, and NaPO_3_ glass leading to the formation of NZSP is taking place effectively. Figure 2d–f shows the XRD patterns of the SSR-NZSP. Under each sintering condition, NZSP was crystallized as the main phase. However, ZrO_2_ impurities were also obtained from every case. Both the samples sintered at 1000 °C and 1100 °C showed that the raw material Na_2_ZrSi_2_O_7_ still remained, suggesting that the solid-state reaction proceeded incompletely. When the sintering temperature rose to 1200 °C, the crystallization peak of Na_2_ZrSi_2_O_7_ disappeared, indicating that the solid phases had reacted completely. Therefore, 1200 °C was considered to be the optimum temperature for the solid-state reaction.

The refined XRD pattern of the LPS-NZSP calcined at 1000 °C for 10 h shows in Figure 3. The content of the NZSP, Na_2_ZrSi_2_O_7_, and ZrO_2_ were analyzed by PROFEX and the result shows in Figure 4. The fitting was successfully converged with R_wp_ = 4.45%. We also listed the refined lattice parameters of LPS-NZSP in Table 1. The crystallite size were all above 100 nm, so there does not seem to be any significant change in crystallinity. With the calcination temperature increased, the content of the NZSP increased simultaneously, and the content of ZrO_2_ decreased Correspondingly. Notablely, the LPS-NZSP calcined at 1000 °C for 10 h showed the content of ZrO_2_ (6%) was on a sharp decrease, and the content of NZSP (74%) increased rapidly. Compared to the SSR-NZSP calcined at 1000 °C for 10 h with the content of ZrO_2_ (15%) and NZSP (57%), LPS-NZSP preferred a superiority. It indicated that NaPO_3_ glass as the additive did work on promoting the solid-state reaction at a lower temperature. Moreover, the decrease of the ZrO_2_ will be hoped to increase the ionic conductivity.

Figure 5 shows the change curves of bulk density and relative density of the calcined LPS-NZSP. For the same calcination time (3, 5, and 7 h), the bulk density and relative density decreased with the calcination temperature increasing. However, 10 h showed a converse trend that the bulk density and relative density increased with the calcination temperature increasing. It can be inferred that the form of the porosity is owing to the proceed of liquid-phase sintering. In a short sintering time, the liquid-phase sintering with a higher sinter temperature proceed faster and it will be more porosity to lead to the bulk densities decrease. However, in an enough sintering time, the higher sinter temperature will synthesize a denser pellet body. Therefore, the NaPO_3_ glass as the liquid phase is conducive to form a denser body at a lower temperature and shorter time.

The LPS-NZSP calcined for 10 h exhibited a lower activation energy was considered that NaPO_3_ glass reduced the grain boundary resistance. Therefore, the SEM cross-section observation of 10 h sintered body was contrasted and shown in Figure 6. From the result of the comparison, the pores of the LPS-NZSP were clearly bigger than that of the SSR-LPS. It can be owing to the escape of the moisture in the sintering process. Despite the hygroscopicity of NaPO_3_ glass reduced the bulk density of the composite electrolytes, the amount of the pores and the porosity also decreased clearly. Moreover, it can be observed that the grain was welded to grain compactly in the microstructure of the LPS-NZSP. This can be attributed to the softened NaPO_3_ glass flowed through the grain boundary to fill in the gap, reduced the grain boundary and promoted the solid-state reaction.

XPS was employed to analysis the content proportion of the elements Na, Zr, Si, P, O. The Na 1s, Zr 3d, Si 2p, P 2p and O 1s XPS spectra collected from the pellet samples of the samples. Figure 7 shows the comparison of survey XPS spectra of the samples sintered at 1000 °C for 10 h. We can observe that the sodium content proportion of the SSR-NZSP is higher than that of the LPS-NZSP. However, calculated from the ratio of Na/Si, both NZSPs exhibited that the Na is excess. It indicated that a part of Na existed in the ceramics surrounded the NZSP crystal. This will be propitious to the conductivity of sodium ion. In addition, calculated from the ratio of Si/P, we can find that the value of the SSR-NZSP was smaller than that of the LPS-NZSP. It means that more excess phosphate existed in the SSR-NZSP and this obviously will block the conduction of Na ion. From this perspective, in the sintering condition of 1000 °C for 10 h, the activation energy of the LPS-NZSP is lower than that of the SSR-NZSP will be explained.

### 3.3. Electrical Conductivity

Figure 8 shows the Nyquist plots for the ionic conductivities of the samples measured from 100 °C to 200 °C. By means of comparing the ionic conductivities between the two kinds of electrolytes under 100 °C, we found that the ionic conductivity of the LPS-NZSP is higher than that of the sample by SSR-NZSP in the sintering condition of 1000 °C for 7 h and 1000 °C for 10 h. Only the LPS-NZSP calcined of 1000 °C for 10 h shows the one semi-circle (108 Ωcm) according to the grain boundary but the other spectra did not show semi-circles. We decided to determine the ionic conductivity from the sum of the resistances of the bulk and grain boundaries. It indicated that use the feature of low melting point of NaPO_3_ glass will improve the process of solid-state reaction under a relatively lower sintering temperature and a longer sintering time.

Arrhenius plot of ionic conductivities of the LPS-NZSP are showed in Figure 9. It shows clearly that the ionic conductivities will increase with the sintering temperature going up. However, the ionic conductivities of the LPS-NZSP calcined for 10 h did not exhibit a superiority with the sintering temperature increasing. It means that the LPS-NZSP calcined at 1000 °C for 10 h showed the same ionic transport ability against the 1100 °C for 10 h and 1200 °C 10 h. This could be also attributable to the NaPO_3_ glass.

To analyze the role of the NaPO_3_ glass, activation energies were calculated via Arrhenius equation based on the fitting slope of Arrhenius plots. Figure 10 shows that the activation energies will increase with the increase of sintering temperature. In addition, though the LPS-NZSP calcined at 1000 °C and 1100 °C showed a near activation energy, the LPS-NZSP calcined at 1000 °C for 10 h exhibited a preponderant low activation energy 0.18 eV, which is 62% lower than the SSR-NZSP 0.29 eV. It indicated that the NaPO_3_ glass will work in reducing the grain barrier in a quite long sintering time and low sintering temperature. On the other side, the sintering time is longer, NaPO_3_ glass will flow and diffuse sufficiently. For this reason, the longest sintering time 10 h showed the lowest activation energy in the different sintering time.

The ceramics sintered at 1000 °C showed favorable electrical conductivity despite the residual Na_2_ZrSi_2_O_7_. The absence of ZrO_2_, which does not contribute to the conductivity of sodium ions, is one of the reasons for the improvement in electrical conductivity, but the presence of Na_2_ZrSi_2_O_7_ should be noted. In the crystal structure of Na_2_ZrSi_2_O_7_, there is a large space that can contain water molecules [40,41,42]. It is curious that the ionic conductivity of Na_2_ZrSi_2_O_7_ has not been evaluated so far, but the results of this study strongly suggest that Na_2_ZrSi_2_O_7_ as well as the NZSP phase contribute to the sodium ion conductivity.

## 4. Conclusions

We synthesized the Na_3_Zr_2_Si_2_PO_12_ ceramics by adding the NaPO_3_ glass successfully. The LPS-NZSP calcined at 1000 °C for 10 h showed a great superiority to synthesize the NZSP. Owing to the hygroscopic of the NaPO_3_ glass, the pores formed in the sintering process of the LPS-NZSP can be observed bigger than that of the samples by SSR. Under the condition of 1000 °C for 10 h, the LPS-NZSP showed the same ionic transport ability against the 1100 °C for 10 h and 1200 °C for 10 h. Furthermore, the LPS-NZSP calcined at 1000 °C for 10 h exhibited a preponderant low activation energy 0.18 eV, which is pretty lower than the SSR-NZSP. This indicated that the NaPO_3_ glass did work on reducing the grain barrier and facilitate the solid-state reaction at the lower sintering temperature and longer sintering time. Moreover, in the sintering at 1000 °C for 10 h, the LPS-NZSP showed the ratio of Si/P is closer to the composition Na_3_Zr_2_Si_2_PO_12_. In summary, the NaPO_3_ glass as the additive to synthesis the NASICON-type Na_3_Zr_2_Si_2_PO_12_ can lower the sintering temperature, improve the solid-state reaction and reduce the grain barrier. It can be expected to be an advantageous synthesis method for the solid oxide electrolyte.

## Figures and Tables

**Figure 1 materials-14-03790-f001:**
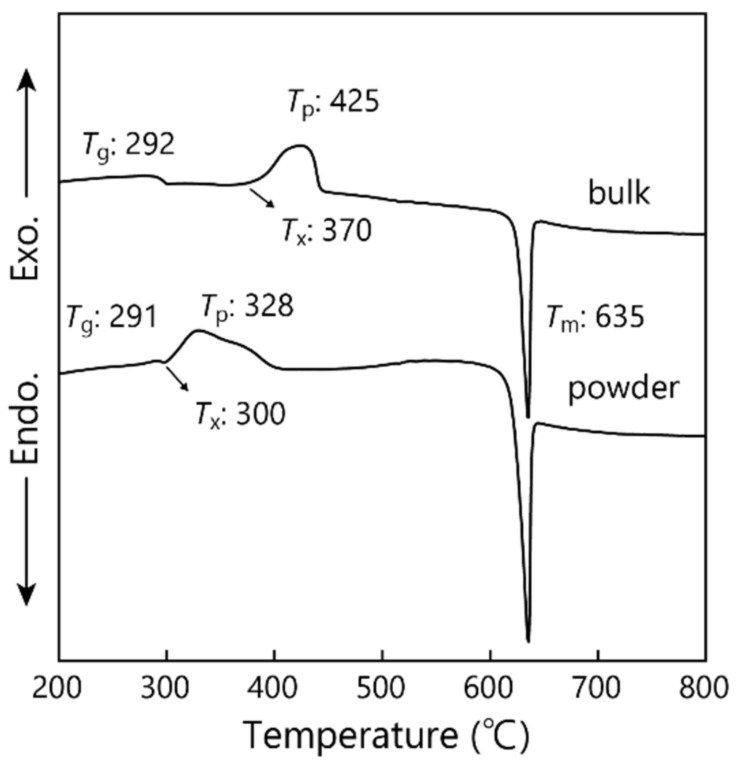
DTA curves of NaPO_3_ glass.

**Figure 2 materials-14-03790-f002:**
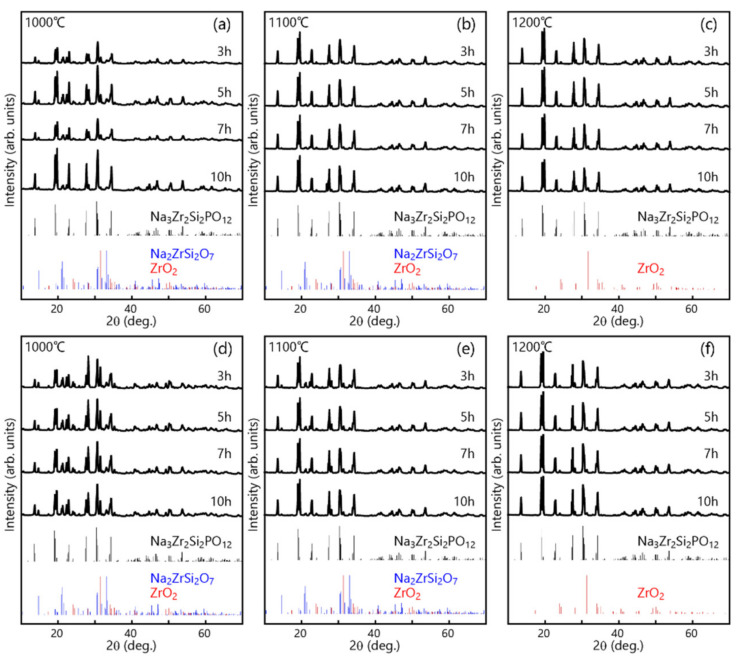
XRD patterns of the LPS-NZSP (**a**–**c**) and SSR-NZSP (**d**–**f**) calcined at various temperature.

**Figure 3 materials-14-03790-f003:**
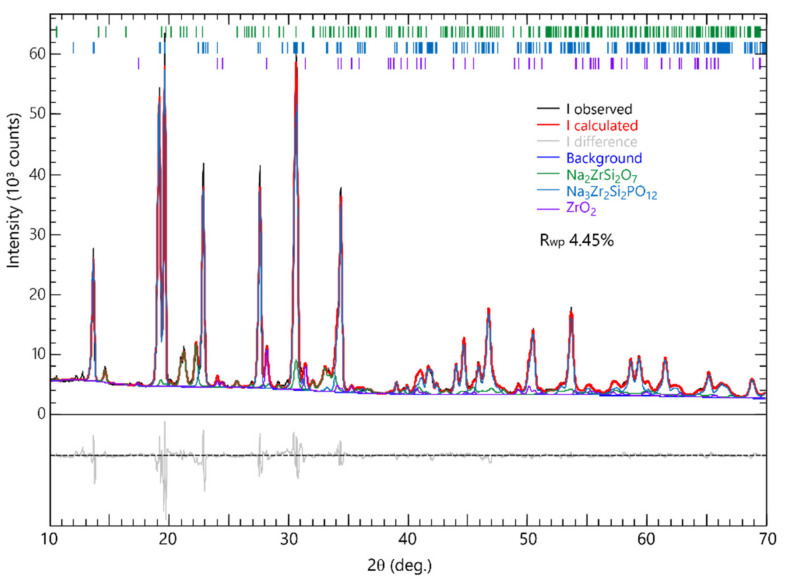
Refined XRD pattern of LPS-NZSP calcined at 1000 °C 10 h.

**Figure 4 materials-14-03790-f004:**
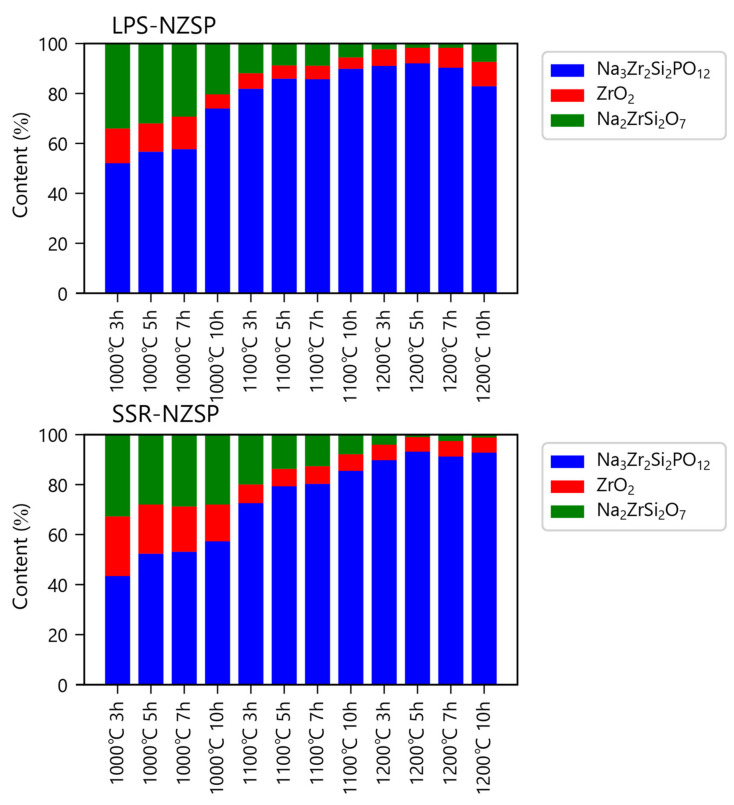
Content of formed phases determined by Rietveld fitting.

**Figure 5 materials-14-03790-f005:**
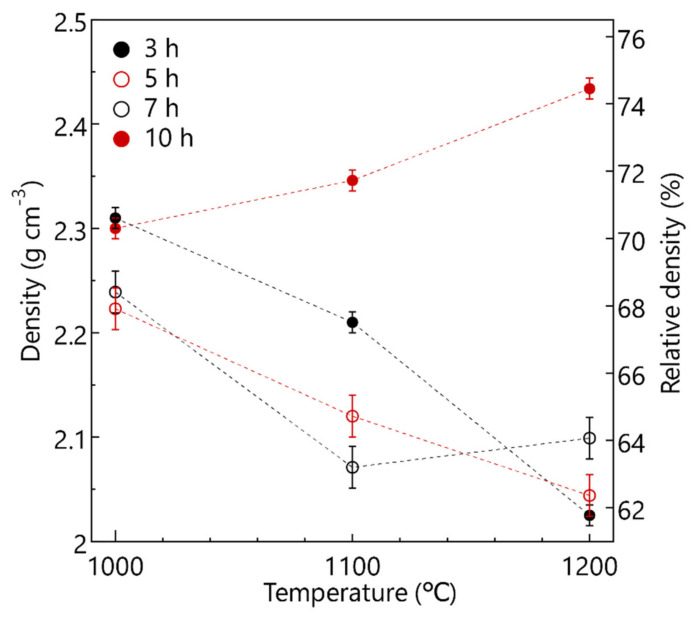
Bulk density and relative density of the LPS-NZSP as a function of calcination temperature.

**Figure 6 materials-14-03790-f006:**
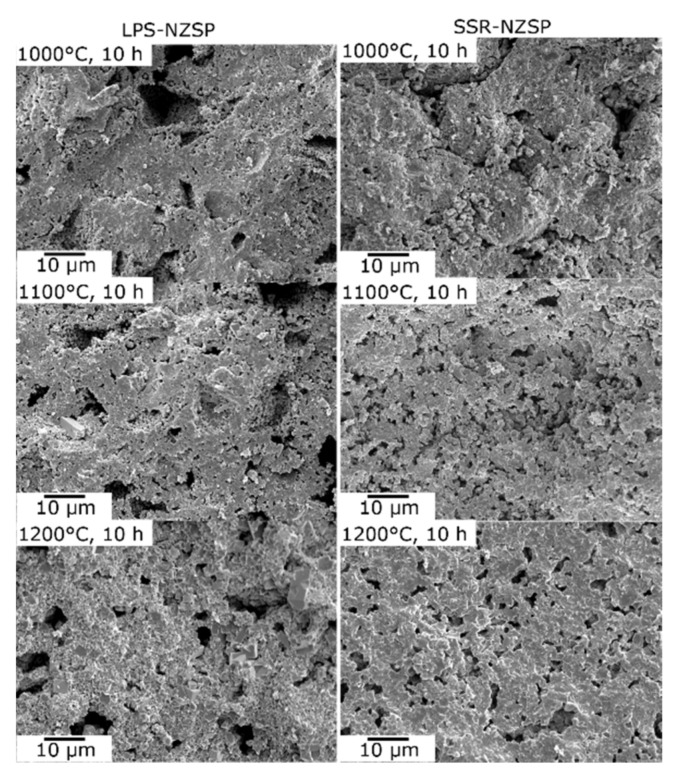
SEM cross-section observation of the samples.

**Figure 7 materials-14-03790-f007:**
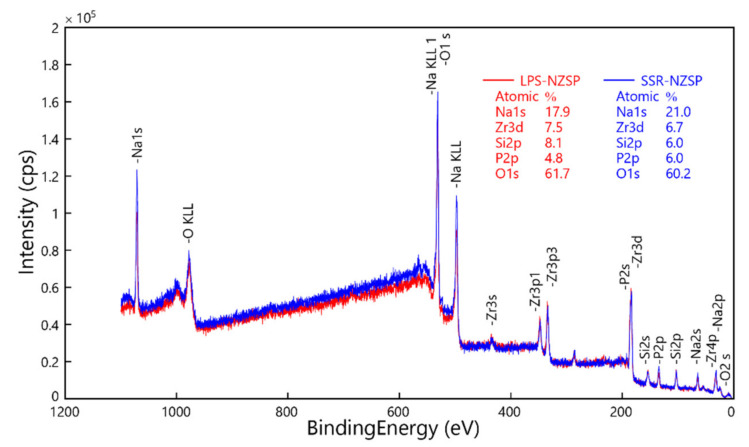
XPS survey spectra of the samples calcined at 1000 °C for 10 h.

**Figure 8 materials-14-03790-f008:**
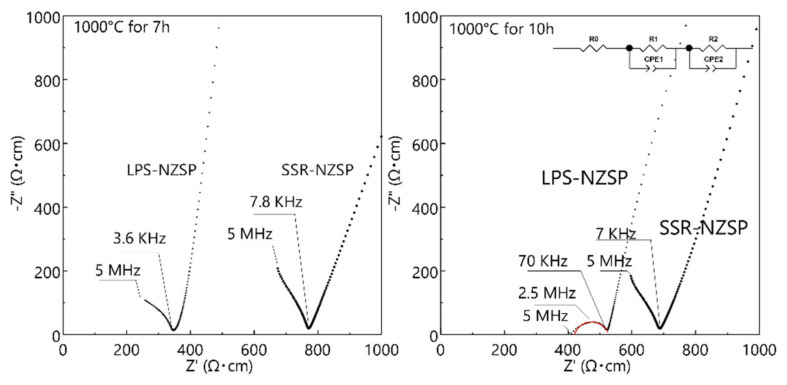
Nyquist plots of the samples calcined at 1000 °C for 7 h and 10 h. The sampling was performed at 100 °C. The red line mean fitted curve using equivalent circuit as shown in the figure.

**Figure 9 materials-14-03790-f009:**
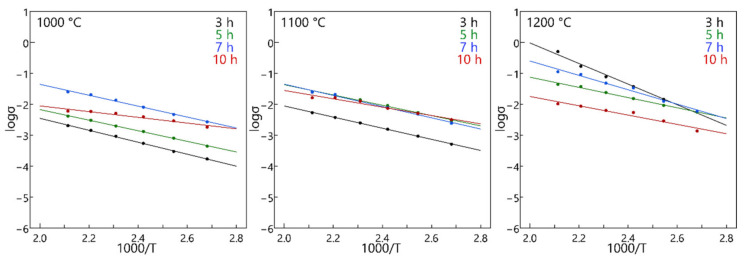
Arrhenius plots for electrical conductivity of the LPS-NZSP.

**Figure 10 materials-14-03790-f010:**
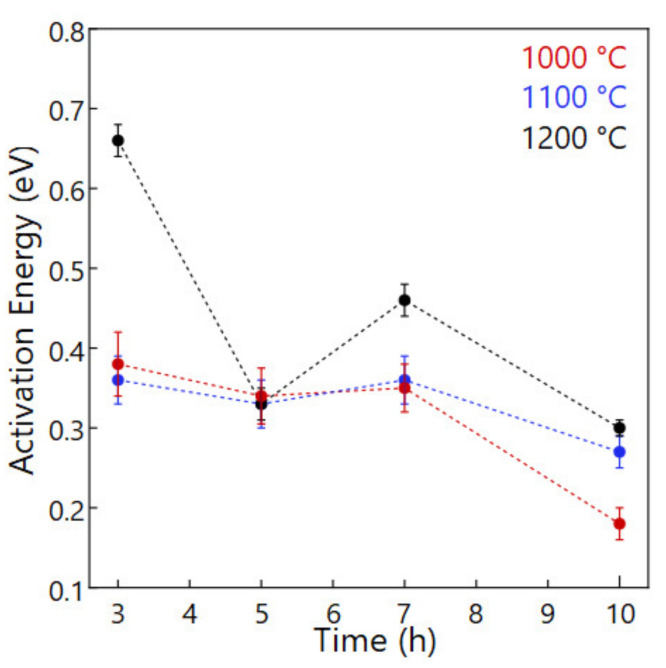
Activation Energy of the LPS-NZSP.

**Table 1 materials-14-03790-t001:** Refined lattice parameters of LPS-NZSP.

Heat Treatment Condition	a (nm)	b (nm)	c (nm)	Beta (Deg.)
1000 °C for 3 h	1.5604 (1)	0.9003 (4)	0.9244 (3)	124.25 (2)
1000 °C for 5 h	1.5617 (1)	0.9015 (2)	0.9247 (3)	124.23 (4)
1000 °C for 7 h	1.5641 (1)	0.9012 (1)	0.9241 (2)	124.20 (4)
1000 °C for 10 h	1.5652 (1)	0.9018 (4)	0.9233 (2)	124.15 (3)
1100 °C for 3 h	1.5647 (4)	0.9043 (4)	0.9227 (4)	123.76 (3)
1100 °C for 5 h	1.5650 (1)	0.9046 (4)	0.9227 (1)	123.74 (3)
1100 °C for 7 h	1.5648 (1)	0.9046 (4)	0.9228 (2)	123.73 (1)
1100 °C for 10 h	1.5653 (1)	0.9055 (4)	0.9220 (2)	123.72 (2)
1200 °C for 3 h	1.5647 (1)	0.9051 (4)	0.9223 (4)	123.69 (3)
1200 °C for 5 h	1.5644 (1)	0.9049 (3)	0.9224 (2)	123.67 (4)
1200 °C for 7 h	1.5642 (1)	0.9047 (3)	0.9224 (3)	123.65 (1)
1200 °C for 10 h	1.5644 (1)	0.9049 (3)	0.9217 (4)	123.68 (2)

## Data Availability

The data that support the findings of this study are available from the corresponding author, T.H., upon reasonable request.

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
