# Peer review of "Synthesis and Na+ Ion Conductivity of Stoichiometric Na3Zr2Si2PO12 by Liquid-Phase Sintering with NaPO3 Glass"

_materials, 2021, doi:10.3390/ma14143790_

Round 1
Reviewer 1 Report
Dear Authors
The paper "Synthesis and Na+ ion conductivity of stoichiometric Na3Zr2Si2PO12 by liquid-phase sintering with NaPO3 glass" seem to be interesting but before final acceptation should be corrected In my opinion the major revision is required.
Please improve the paper by ponit by point
1.Electrical circuit should be added and analysed with electrical results obtained from ac impedance spectroscopy
2. Relative density of sintered samples should be analysed vs. temperature of sintering samples
3. Please mark the frequency on the ImZ-ReZ
Author Response
- Electrical circuit should be added and analysed with electrical results obtained from ac impedance spectroscopy
We added the frequency, equivalent circuit, and fitting curves in Fig8. Only LPS-NZSP at 1000°C for 10h shows a semicircle, which is supposed to be the grain boundary resistance. The other Nyquist plots show only part of the semicircle. A detailed explanation has been added in Section 3.3.
- Relative density of sintered samples should be analysed vs. temperature of sintering samples
The relative density is shown in the second axis of Fig. 5 with the density of NZSP as 3.26g/cm3.
- Please mark the frequency on the ImZ-ReZ
We have added in some of the frequencies in revised Fig. 8.
Reviewer 2 Report
The paper represents the study of synthesis of NZSP uisng NaPO3 glass as an sintering/ liquid aid. The study is well organized, however I have some comments which should be discussed/added in a final version of the paper, before it shoul dbe accepted by "Materials".
Line 74: was there a waiting time between milling to avoid extensive heat development, which liqiud was used during milling ?
line 78: why 32 mikrometer particle size, please explain, and how this was controlled ?!
line 106ff: why was the condictivity beasured between 100°c and 200°C, what is about RT, whould not more data yile dbetter estimates of activatin energies
126: Na2ZrSi2O7 precursor: why using this precursor, are there any advantages when using this instead of oxidic starting material, What about the phase purity of the precursor.
XRD pattern: You wrote about an increase of intenisty of peaks belonging to NZSP, is this just an effect of scale factors or is there also an increase in line width, i.e. changes in crystallite size. Generally, since authors use profex Rietveld refinements, comment of size development from microstructure analysis, i.e. is there any change in peak width with sintering time and temperature.
How do the lattice parameters evolve, are there any changes ? Often small, but significant changes are observed, i.e. decreasing unit cells with increasing crystallinity
Line 143: use of word "percipitation": i would prefer the term crystalization, one precipitates a phase from a solution...
line 157: "correspondingly" small case
line 155: I do not understand that statement,
Fig 4 shows the phase evolution of several samples, not only 1000°C
Fig 3: difference plot looks quite "bad", have you refined lattice parameters?
Figure 4: estimation of error bars (estimated stanard deviation of phase content) in the figure would be fine, can you include data for the solid state sintering also sin some way, to see the superiority f NaPO3 aided synthesis ?
Figure 4 : comment on the fact that ZRSP content decreases at 1200°C with prolonged sintering time, loss of sodium ? So would not be 1150°C a good choice, please comment on this.
Generally, it seems that the ZrO2 content is lowest at 1100°C, is there a possibility to reduce ZrO2 further ? Seems that this is the most critical phase reducing overall conductivity of the pellets.
Add error bars to Figure 5
line 175: can you comment on the densites in more detail, so if T increases, more porous pellets ae obtaind, expect at 1200°C ? Explain.
XPS: only collected for one sample ? Evolution of composition with T and time would be very interetsing. Comparing the LPS and SSR samples, is this just an effect of different starting materials/ wightes amounts, or indeed is there an effect of synthesis method, please comment on this as this is an essential infromation.
Figure 5: can you estimate error bars for values ? Data for 5 hours at 1200°C are off the trend, have you tried to repeat the measurement ? Or what is different/ wrong with this sample.
Fig 10: these are all the LPS samples ? Please note this, same for othere figures...
Author Response
Line 74: was there a waiting time between milling to avoid extensive heat development, which liqiud was used during milling ?
The waiting time is 10 minutes and the liquid is acetone, which has been added in the manuscript.
line 78: why 32 mikrometer particle size, please explain, and how this was controlled ?!
A sieve was used to obtain a powder of less than 32 micrometers. Descriptions have been added to the revised manuscript.
line 106ff: why was the condictivity beasured between 100°c and 200°C, what is about RT, whould not more data yile dbetter estimates of activatin energies
Because we measured the conductivity in the air and the moisture in the air can influence on the temperature dependency. So in order to remove the influence made by the moisture, we measured the conductivity between 100°C and 200°C.
126: Na2ZrSi2O7 precursor: why using this precursor, are there any advantages when using this instead of oxidic starting material, What about the phase purity of the precursor.
We chose the NaPO3 glass as a partial component of the NZSP, hoping to reduce the grain boundary resistance. In order to synthesize the stoichiometric NZSP with NaPO3 glass, the precursor Na2ZrSi2O7 is necessary.
The precursor Na2ZrSi2O7-ZrO2 composite powder contained 35%ZrO2 and 65%Na2ZrSi2O7.
XRD pattern: You wrote about an increase of intenisty of peaks belonging to NZSP, is this just an effect of scale factors or is there also an increase in line width, i.e. changes in crystallite size. Generally, since authors use profex Rietveld refinements, comment of size development from microstructure analysis, i.e. is there any change in peak width with sintering time and temperature. How do the lattice parameters evolve, are there any changes? Often small, but significant changes are observed, i.e. decreasing unit cells with increasing crystallinity
The lattice parameters are listed in Table1 in revised manuscript. The crystallite size were estimated all above 100 nm by PROFEX, so there does not seem to be any significant change in crystallinity.
Line 143: use of word "percipitation": i would prefer the term crystalization, one precipitates a phase from a solution...
The word has been revised.
line 157: "correspondingly" small case
line 155: I do not understand that statement,
Fig 4 shows the phase evolution of several samples, not only 1000°C
The statement has been revised.
Fig 3: difference plot looks quite "bad", have you refined lattice parameters?
Thank you for your suggestion. We tried refinement again and refinement was successfully converged. The results are shown in Fig.4 and related sentence are added in the revised manuscript.
Figure 4: estimation of error bars (estimated stanard deviation of phase content) in the figure would be fine, can you include data for the solid state sintering also sin some way, to see the superiority f NaPO3 aided synthesis ?
We added the content of phases in SSR-NZSP in revised manuscript. The compare of the LPS-NZSP and SSR-NZSP is showing below. It shows clearly that the ZrO2 content of LPS-NZSP is absolutely lower than that of SSR-NZSP sintering at 1000°C. So the NaPO3 glass is contribute to the solid-state reaction.
Figure 4 : comment on the fact that ZRSP content decreases at 1200°C with prolonged sintering time, loss of sodium ? So would not be 1150°C a good choice, please comment on this.
With the sintering time prolonged, the sodium will be anticipated to volatilize and the content of NZSP will decrease correspondingly. If the 1150°C was chose, the temperature difference will be different. And it can’t be contrast to 1000°C and explain the content of NZSP correctly.
Generally, it seems that the ZrO2 content is lowest at 1100°C, is there a possibility to reduce ZrO2 further ? Seems that this is the most critical phase reducing overall conductivity of the pellets.
Yes, reduction of the content of ZrO2 is the key to increase the ionic conductivity. We designed this sintering method to synthesize the NZSP, and had tried our best to reduce the ZrO2.
Add error bars to Figure 5
The error bars has been added.
line 175: can you comment on the densites in more detail, so if T increases, more porous pellets ae obtaind, expect at 1200°C ? Explain.
It can be inferred that the form of the porosity is owing to the proceed of liquid-phase sintering. In a short sintering time, the liquid-phase sintering with a higher sinter temperature proceed faster and it will be more porosity to lead to the bulk densities decrease. However, in an enough sintering time, the higher sinter temperature will synthesize a denser sintered body.
XPS: only collected for one sample ? Evolution of composition with T and time would be very interetsing. Comparing the LPS and SSR samples, is this just an effect of different starting materials/ wightes amounts, or indeed is there an effect of synthesis method, please comment on this as this is an essential infromation.
We evaluated the composition of LPS and SSR samples sintered in 1000°C 10h to further confirm that the NaPO3 glass did promote the synthesis of NZSP.
Because all the raw material were weighted in the atomic ratio Na:Zr:Si:P=3:2:2:1, so the synthesis method indeed effect the atomic percent.
Figure 5: can you estimate error bars for values ? Data for 5 hours at 1200°C are off the trend, have you tried to repeat the measurement ? Or what is different/ wrong with this sample.
Yes, we had repeated the measurement but I got the same results. It maybe owing to the sinter temperature is high enough, so the solid-state reaction can proceed steadily and the bulk density show a regularly increase.
Fig 10: these are all the LPS samples ? Please note this, same for othere figures...
The name of the figure 10 has been revised.
Reviewer 3 Report
The authors should consider the following points in revising their manuscript:
(1) This reviewer does not think it is obvious that "plastic deformation cannot be expected in oxide with high Young's moduli" (p. 2). The authors may explain why it is "obvious" or why is this sentence necessary.
(2) The authors should correct the error in describing the XRD measurements "the scanning speed was 5°C/min and the diffraction angle was...."
(3) The authors should describe in detail the sample polishing method in the sentence "...the surface (the thickness of 0.1 mm) of the samples was polished to remove contamination by carbon dioxide in the air." (p. 3).
(4) The authors should report the thickness of the sputtered gold (p. 3).
(5) The authors should explain the non-monotonic dependence of density vs. time for 1000 and 1100 C treated samples in Fig. 5.
(6) What is "... will be got an explanation" (p. 6)?
(7) Is NaPO3 ionically conducting? What is its conductivity? What is the mechanism for NaPO3 to lower the grain boundary barrier?
(8) Please replace "pretty lower" with a more quantitative description.
(9) Please add error bars to Fig. 10. How many repeated measurements did the authors perform to produce the results in Fig. 10?
(10) References 39, 40, 41, and 42 are missing.
Author Response
(1) This reviewer does not think it is obvious that "plastic deformation
cannot be expected in oxide with high Young's moduli" (p. 2). The
authors may explain why it is "obvious" or why is this sentence necessary.
Because the low Young's modulus of sulfide solid electrolytes allows them to deform plastically only under pressure at room temperature, thus suppressing the electrical resistance between particles. The high Young's modulus of solid oxide electrolytes cannot be expected to reduce electrical resistance through plastic deformation. The electrical resistance of solid oxide electrolytes needs to be reduced by the sintering process.
(2) The authors should correct the error in describing the XRD
measurements "the scanning speed was 5°C/min and the diffraction angle
was...."
The diffraction angle was 2q=10-70°. I think maybe the manuscript has some problems. It should be showed in the manuscript but it didn’t.
(3) The authors should describe in detail the sample polishing method in
the sentence "...the surface (the thickness of 0.1 mm) of the samples
was polished to remove contamination by carbon dioxide in the air." (p. 3).
The sample was polished by using diamond wheel. The process also done in Ar filled grove box. The explanation was added in the revised manuscript.
(4) The authors should report the thickness of the sputtered gold (p. 3).
The thickness of the sputtered gold was 10nm. And the message has been added into the manuscript.
(5) The authors should explain the non-monotonic dependence of density
vs. time for 1000 and 1100 C treated samples in Fig. 5.
It can be inferred that the form of the porosity is owing to the proceed of liquid-phase sintering. In a short sintering time, the liquid-phase sintering with a higher sinter temperature proceed faster and it will be more porosity to lead to the bulk densities decrease. However, in an enough sintering time, the higher sinter temperature will synthesize a denser pellet body.
And it has been added into the manuscript.
(6) What is "... will be got an explanation" (p. 6)?
Because the more excess phosphate existed in the SSR-NZSP was conjectured to block the conduction of Na ion. And this will lead to a higher activation energy.
(7) Is NaPO3 ionically conducting? What is its conductivity? What is the
mechanism for NaPO3 to lower the grain boundary barrier?
Yes the NaPO3 is the ionically conducting. Its conductivity is 5.39×10-5 S cm-1 at 230°C. In the sintering process, the NaPO3 glass will melt as a liquid phase over the melting point 635°C. And the NaPO3 melt will fill the gap between the grains. In this way, the NaPO3 glass can low the grain boundary barrier.
(8) Please replace "pretty lower" with a more quantitative description.
“pretty lower” was changed to “62% lower”.
(9) Please add error bars to Fig. 10. How many repeated measurements did
the authors perform to produce the results in Fig. 10?
The error bars were added. The measurements were repeated for three times.
(10) References 39, 40, 41, and 42 are missing.
We apologize the citations had disappeared. Three articles were added as references from 39 to 41 in revised manuscript.
Round 2
Reviewer 1 Report
In my opinion the paper can be accepted for publication in Materials
Author Response
We would like to thank you for your favorable comments.
Reviewer 2 Report
Querries have been answered almost satisfactorily
Author Response

(The authors gave the same response as above.)

Reviewer 3 Report
I am satisfied with the revisions.
Author Response

(The authors gave the same response as above.)
